# Optimization and Modeling of Slightly Acidic Electrolyzed Water for the Clean-in-Place Process in Milking Systems

**DOI:** 10.3390/foods9111685

**Published:** 2020-11-18

**Authors:** Yu Liu, Chaoyuan Wang, Zhengxiang Shi, Baoming Li

**Affiliations:** 1Department of Agricultural Structure and Bioenvironmental Engineering, College of Water Resources and Civil Engineering, China Agricultural University, Beijing 100083, China; xiaohaizhibei@cau.edu.cn (Y.L.); shizhx@cau.edu.cn (Z.S.); libm@cau.edu.cn (B.L.); 2Key Laboratory of Agricultural Engineering in Structure and Environment, Ministry of Agriculture and Rural Affairs, Beijing 100083, China

**Keywords:** slightly acidic electrolyzed water, response surface model, milking system, sanitation

## Abstract

To find an environmentally friendly and energy efficient alternative to acidic detergent for a milking system clean-in-place (CIP) process, this study investigated the feasibility of applying slightly acidic electrolyzed water (SAEW) alone to wash the system by cleaning soiled stainless steel (304) pipes, rubber gaskets, and PVC milk hoses, which were used in the milking system. The results showed that SAEW with appropriate parameters could achieve the same or even better hygienic effects compared with commercial detergent. Using response surface models, the SAEW parameters required to clean stainless steel were optimized at 9.9 min for the treatment time, 37.8 °C for the water temperature, and 60 mg/L for the available chlorine concentration; and were 14.4 min, 29.6 °C, and 60 mg/L for rubber gasket and PVC samples, respectively. After washing with the optimized parameter combination, bacteria and adenosine triphosphate on the three materials were almost non-detectable, indicating that SAEW has the potential to replace acidic detergents in CIP milking systems.

## 1. Introduction

Cleaning and disinfection are critical operations in dairy farming. On a commercial dairy farm, the clean-in-place (CIP) process is a standard process of the milking system, consisting of four cycles: (1) prewashing with warm water; (2) circular washing with alkaline chemical detergent; (3) circular washing with acidic chemical detergent; (4) sanitizing with sanitizing detergent before the next milking phase [1]. The commercially used detergents in the CIP process typically contain sodium hydroxide, sulfuric acid, or phosphoric acid, which are highly caustic and can cause serious burns to the skin and mucous membrane [2,3,4,5]. Moreover, detergent residues in raw milk seriously affect the milk quality and reputation of the farms, which consequently lead to heavy economic losses to the farmers [6].

Under such circumstances, research endeavors have been conducting to optimize the CIP protocols and find alternative detergents. Studies have shown that the optimization of cleaning protocols, aimed at minimizing the consumption of energy and chemical detergent, depends on the type of soils, surfaces, and materials to be cleaned [7]. For the milking system, Vilar et al. [8] found that weekly cleaning with acidic chemical detergent achieved a better hygienic environment than daily cleaning. Liu et al. [9] suggested that using alkaline and acidic detergent alternately to clean the milk tank could help maintain a level of good hygiene and save water and costs. Overall, it is not suggested to apply alkaline and acidic chemical detergents together in one cleaning procedure, especially for equipment with clod surfaces, such as milking systems, because of the associated corrosivity, energy use, and complexity of the discharged water treatment afterward [10]. Therefore, finding an alternative that is suitable for interchangeable use to replace these chemicals on sites would be highly advantageous to farmers.

Electrolyzed water (EW), as a novel cleaning and sanitizing agent, has been widely applied in food and medical industries [11,12]. EW can be disposed back into the environment after usage, contributing to decreasing the amount of halogenated compounds that accumulate in the environment, as highlighted by the U.S. Environmental Protection Agency [13]. An EW generator with a semipermeable membrane can produce both alkaline electrolyzed water (pH of 11.5) and acidic electrolyzed water (AEW, pH of 2.6) at the cathode and anode in the same process, respectively [14], and the characteristics of both solutions theoretically satisfy the requirements for alkaline and acidic chemical detergents in cleaning the milking system [2]. Studies in the laboratory have shown that using alkaline electrolyzed water (pH of 11.5) and AEW (pH of 2.6) together in a complete CIP process could achieve an equivalent or even better cleaning effect [2,5,15,16,17]. However, the practical challenge of using alkaline electrolyzed water and AEW together is the higher corrosivity. To optimize this protocol, Liu et al. [18] showed that alkaline electrolyzed water as an alternative to alkaline chemical detergents could be used alone in standardized CIP processes for milking systems.

Meanwhile, an EW generator without a semipermeable membrane produced slightly acidic electrolyzed water (SAEW, pH of 5.0–6.5) in one study [19]. Another study showed that AEW corrosivity decreased by more than 50%, with the pH increasing from 2.42 to 6.12 [20], and the strong oxidability and good sterilization ability of the SAEW were also revealed [21,22,23]. Additionally, the consumption of energy to produce SAEW is 50% less than that needed to produce the same quantity of AEW due to the different membrane structures. Theoretically, SAEW meets the standards for an acidic detergent alternative in the CIP process, however its effectiveness in cleaning milking systems has not yet been tested. The chemicals used in the CIP process are toxic, dangerous, and expensive, while using SAEW will help eliminate many of the dangers and will be cost-effective. The successful application of SAEW alone to wash milking systems would greatly benefit the optimization of the CIP cleaning protocols.

This work aimed to investigate the cleaning effect of using SAEW alone on three typical materials (stainless steel (304) pipes, rubber gaskets, and PVC milk hoses) that are often used in milking systems and to optimize the combinations of key parameters (treatment time, water temperature, and available chlorine concentration (ACC)) of the SAEW cleaning through response surface modeling.

## 2. Materials and Methods

### 2.1. Bacterial Cultures

This study selected *Escherichia coli* (ATCC25922) and *Pseudomonas fluorescens* (ATCC49642) as target bacteria, which can be easily found in raw milk. These two bacteria were collected from the China Veterinary Culture Collection (CVCC, Beijing, China and were cultured in 10 mL of tryptic soy broth (TSB, AOBOX Biotechnology Co. Ltd., Beijing, China) at 37 °C for 24 h. To ensure the culture broth had a bacterial population of 1 × 10^9^ CFU/mL, each sample of the broth was checked using an aerobic plate count (APC) method with tryptic soy agar (TSA, AOBOX Biotechnology Co. Ltd., Beijing, China) [15].

### 2.2. Milk Preparation

Raw milk was obtained from the Yulong dairy farm (Beijing, China). To increase the raw milk bacterial population, 1 mL of each culture broth was centrifuged at 4400× *g* for 4 min at 4 °C [15]. Two bacterial cells were resuspended together in 10 mL of raw milk with a mixer (WH-2, Shanghai Luxi Fenxiyiqi Co. Ltd., Shanghai, China).

### 2.3. Specimens Preparation

Three typical materials used in milking systems, namely stainless steel (304) pipes, rubber gaskets, and PVC milk hoses, were prepared as specimens. For each material there were 12 specimens, which were cut into 10 cm^2^ pieces. Before every treatment, the specimens were cleaned and autoclaved at 121 °C for 20 min.

To prepare contaminated specimens, 0.1 mL inoculated raw milk was evenly soiled on the whole surface of each specimen with a sterile glass-coated rod. Then, the specimens were dried using laminar flow for 2 h to evaporate all visible liquid. The initial concentrations of bacteria on stainless steel, rubber gasket, and PVC samples were 5.10–6.00 log_10_ CFU/cm^2^, 5.23–5.94 log_10_ CFU/cm^2^, and 3.00–5.64 log_10_ CFU/cm^2^, respectively.

### 2.4. Response Surface Design and Validation

A standard Box–Benken response surface design (Table 1) was utilized to select suitable combinations of treatment time, water temperature, and ACC for SAEW cleaning, which contained 15 experimental trials, with the center point experiment repeated three times. According to the results of pre-experiments, the ranges for treatment time (8–12 min), water temperature (20–40 °C), and ACC (40–60 mg/L) for stainless steel pipe, and the ranges of treatment time (12–15 min), water temperature (20–30 °C), and ACC (40–60 mg/L) for rubber gasket and PVC milk hose samples were determined. For each trial three replications were performed.

An additional 6 trials with different parameters, which were not included in the model development, were carried out (Table 2) to confirm the adequacy of the models.

### 2.5. Preparation of SAEW and Chemical Disinfectant

SAEW samples with different ACC levels of 20, 30, 40, 50, and 60 mg/L were generated by electrolyzing NaCl solutions using a generator (Beijing Rui’ande Environment Technology Co. Ltd., Beijing, China) set at 5, 8, 11, 14, and 17 A, respectively. The pH and oxidation–reduction potential (ORP) values of the SAEW were measured using a dual-scale meter (Hangzhou Ying’ao Technology Co., Hangzhou, China). The ACC was determined by a digital chlorine test kit (RC-2Z, Kasahara Chemical Instruments Co., Saitama, Japan). The chemical disinfectant was prepared at a concentration of 0.5% by diluting the concentrated acidic chemical detergent (Cidmax, DeLaval Co. Ltd., Tianjin, China; referred to as “super”) with tap water following the manufacturer’s instruction.

The SAEW and commercial acidic chemical detergent were heated in buckets with lids to the targeted temperature using a self-heating plate controlled by a single-channel thermal table (CH6E07, Beijing Kunlun Zhicheng Sensor Technology Co., Beijing, China).

### 2.6. Disinfectant Treatments

In order to check the cleaning potential when using SAEW alone, this study only performed the first and third steps of the standard CIP process (warm water rinsing and acidic chemical detergent washing) without disturbance from the final disinfection step. The cleaning process consisted of two cycles: (1) prewashing with 45 °C warm water for 5 min; and (2) washing with SAEW or “super”.

The cleaning solution was poured into a 400 mL beaker without any agitation to simulate the worst case scenario for locations of slow flow in the milking system [15]. All prepared specimens were soaked in warm water, then transferred to the beakers containing SAEW or “super”. The treatment temperature and time for “super” were 80 °C and 8 min, respectively, according to the manufacturer’s recommendation. After treatment, specimens were taken out and sampled immediately.

### 2.7. Bacterial Counting and Cleanliness Evaluation

The soiling and hygiene levels of the specimens were evaluated before and after treatment with SAEW using the APC method and an adenosine triphosphate (ATP) bioluminescence test. The APC method is widely used to assess the cleanliness of food contact surfaces [8]. The ATP bioluminescence test is well suited to monitoring the cleanliness within hazard analysis critical control point (HACCP) systems [24]. It can be used for real-time assessment and can detect bacterial cells and food residues, which might affect the cleanliness and hygiene of food contact surfaces [25].

Six specimens of each material were checked for their original contamination levels, while the other six specimens were checked for effects after cleaning. Three specimens were swabbed for microbiological analysis using sterilized cotton swabs soaked with 0.1% peptone water, while the other three were swabbed for ATP bioluminescence testing using the method recommended by the Ministry of Health, China [26]. The details for the microbiology counting process were given by Liu et al. [18].

As for the ATP bioluminescence test, samples were collected by using ATP test swabs (Shandong Langrun Commerce CO. Ltd., Jinan, China). Then, the swabs were tested using a LUMinator-T portable analyzer (CF-420, Shanghai Canfu Jidian Co. Ltd., Shanghai, China) to detect the emitted light from the ATP, which was quantified in “relative light units” (RLUs).

The bacteria and ATP removal rates were used for CIP performance comparisons. The equations used to calculate the bacteria and ATP removal rates were as follows:(1)$$Bacteria\;removal\;rate=\left(1-\frac{Bacteria\;concentration\;after\;CIP}{Bacteria\;concentration\;before\;CIP}\right)\times 100\;$$
where the units for the bacteria removal rate and bacterial concentration were % and log_10_ CFU/cm^2^, respectively.
(2)$$ATP\;removal\;rate=\left(1-\frac{ATP\;value\;after\;CIP}{ATP\;value\;before\;CIP}\right)\times 100\;$$
where the units for the ATP removal rate and ATP value were % and RLU, respectively.

### 2.8. Statistical Analysis

The bacteria population was expressed as log_10_ CFU/cm^2^. The mean values for the total aerobic bacteria and RLUs were calculated from the independent triplicate trials. A Box–Benken response surface design table was generated and the results were analyzed using Minitab 17 (Minitab, Inc., State College, PA, USA). Significant differences in mean values for bacteria removal and ATP from three materials were analyzed using least significant differences with repeated measurement analyses of variance (ANOVAs) and a 95% confidence interval in SPSS 21.0 (SPSS, Inc., Chicago, IL, USA).

## 3. Results and Discussion

### 3.1. Physicochemical Properties of Treatment Solutions

The ACC, pH, and ORP values for the tap water, SAEW, and “super” are shown in Table 3. For SAEW, the ACC and ORP values ranged from 18 to 62 and from 732 to 924 mV, respectively. “Super” did not contain any ACC, because its effective elements were H_2_SO_4_ and H_3_PO_4_, and the pH value was 1.61.

### 3.2. SAEW Cleaning Efficiency in Removing Bacteria and ATP

Table 4 illustrates that the disinfection efficacy values for SAEW with different cleaning parameters combinations varied a lot. It can be seen that in some conditions (trials 1, 4, 5, 7, 8, 9, and 15 for stainless steel; trials 5, 7, 11, 12, 14, and 15 for rubber gaskets; trials 1, 2, 4, 5, 6, 7, 9, and 11 for PVC), SAEW treatments had similar or significantly higher (*p* < 0.05) bacterial disinfection efficiency as “super”, as shown by the ATP values and bacteria counts (Table 4). These results indicated that SAEW used with appropriate parameter combinations for treatment time, water temperature, and ACC has the potential to be a cleaning agent for milking systems. The bacteria removal rates for stainless steel, rubber gasket, and PVC samples after using “super” were 99.98%, 100.00%, and 99.83%; and ATP removal rates after using “super” were 69.70%, 93.07%, and 94.04%, respectively. Using SAEW with different combinations of treatment time, cleaning temperature, and ACC gave 100.00% bacterial and ATP removal rates. Bremer et al. [27] and Parkar et al. [28] also stated that the cleaning efficacy of CIP systems significantly depends on the exposure time, temperature, and cleaning agent concentration. Meanwhile, three materials treated with SAEW achieved a 5-log reduction in bacterial species, which corresponded with the definition of sanitization recommended by the Food and Drug Administration [29].

SAEW shows strong oxidability with effective elements (ACC) of HClO, ClO^−^, and Cl_2_. HClO can kill bacteria by destroying the membrane, leading to leakage of the cytoplasmic content [30], protein denaturation, and stopping cellular metabolism [31]. Similarly, the disinfection efficiency of “super” is based on H_2_SO_4_ and H_3_PO_4_, causing protein denaturation in the bacterial cell wall. Although the bactericidal substances of the two cleaning solutions are different, the sterilizing mechanisms are similar. Hence, with appropriate selection of its parameters, SAEW could be used as an alternative to “super”.

### 3.3. Model Fitting

Response surface models were used to predict log_10_ bacterial reductions and ATP removal rates from the milking system and to determine the parameters based on a predicted result. The coefficient of determination (R^2^) and lack of fit values for the 6 models are listed in Table 5 for log_10_ bacterial reduction and ATP removal rates, respectively. The significance of the lack of fit test shows the quality and accuracy of the fitness models; when the *p*-value is greater than 0.05, the model is considered logical [32]. According to the regression analysis of all models, the significance levels of the models were <0.05, while the significance levels for lack of fit were >0.05, which showed that all models were logical.

Among these models, ACC only had a significant and positive effect on log_10_ bacterial reduction for stainless steel (*p* < 0.05). Moreover, all models were greatly affected by treatment time and temperature with SAEW (*p* < 0.05). SAEW at an ACC of 40 mg/L reduced the *Escherichia coli* with 0.3% bovine serum albumin to undetectable levels after 10 min treatment, and it was reduced to an undetectable level with SAEW at an ACC level of 60 mg/L after 5 min treatment [33]. In this study, two bacteria (*Escherichia coli* and *Pseudomonas fluorescens*) were inoculated into the raw milk, and the lowest treatment time and ACC set for cleaning stainless steel were 8 min and 40 mg/L, respectively, explaining why more time and higher ACC were needed to achieve satisfactory hygiene levels. As for rubber gasket and PVC samples, the treatment time (12–15 min) was enough to diminish the leading role of the ACC. Thus, the determination of the three parameters of SAEW is very crucial to achieve a more satisfactory CIP performance because of the interaction effects of treatment time, water temperature, and ACC.

Davidson et al. [34] illustrated that ATP bioluminescence techniques were more sensitive than traditional plate counts for determining surface hygiene. The response surface plots for stainless steel, rubber gasket, and PVC ATP removal rates showed the cleaning effects of treatment time and cleaning temperature (Figure 1).

From the trends of the plots and the statistical analyses, the treatment time and cleaning temperature of SAEW within a certain interval significantly and positively (*p* < 0.05) affected the ATP removal rate for the three materials. With the treatment time increasing, the SAEW water flow could remove deposits more easily. Most of the minerals contained in the milk were acid-soluble [16], which could be removed more effectively with a long SAEW treatment. Water temperature affects the removal of bacteria and the ATP response within a certain range. Increasing the water temperature would enhance the molecular kinetic energy of the ACC to increase the contact opportunity with soils, which could remove more contaminants, as proven by Walker et al. [5,15]. The increase in temperature would result in more ACC being released to air, thus decreasing the sterilization effect of SAEW, especially when the temperature is over 45 °C [2].

### 3.4. Validation of the Models

To validate the models, six additional random trials (Table 2) were carried out. The selected parameters were a treatment time range of 8 to 12 min, temperature range of 20 to 40 °C, and ACC range of 40 to 60 mg/L for stainless steel. Rubber gasket and PVC samples had a treatment time range of 12 to 15 min, temperature range of 20 to 30 °C, and ACC range of 40 to 60 mg/L. The results matched with the predicted results (Figure 2). The R^2^ values of log_10_ bacterial reduction models for stainless steel, rubber gasket, and PVC samples were 0.98, 0.93, 0.96; and the R^2^ values for the ATP removal rate models for the three materials were 0.91, 0.91, and 0.99, respectively. It was concluded that the models were able to reliably predict log_10_ bacterial reduction and ATP removal rates for the three typical materials used in the milking system with SAEW treatment.

### 3.5. Optimization and Validation of SAEW Cleaning Parameters

With the developed models, the SAEW parameters used to clean the milking system could be predicted and optimized to achieve 7.0 log_10_ CFU/cm^2^ bacterial reduction and 100% ATP removal rate on the surfaces (Table 6). The optimized SAEW treatment time, cleaning temperature, and ACC needed to clean stainless steel were 9.9 min, 37.8 °C, and 60 mg/L, respectively, similar to the parameters recommended by Dev et al. [2], who gave an optimized cleaning time of 10 min and temperature of 39.8 °C for AEW (pH = 2.6, ACC = 80 ppm) in order to sanitize a stainless steel pipe in a milking system, however the corrosivity of AEW is 2 times greater than that of SAEW [20], which is believed to decrease the life of a milking system. Additionally, it seems that the bactericidal capability of SAEW is also higher than that of AEW because of its lower cleaning temperature and ACC. Stainless steel is easier to clean than rubber and PVC [16]. It can be seen in Table 6 that the treatment times for rubber gasket and PVC milk hose samples were longer than those for stainless steel pipe samples. Rubber has microscopic caverns and crevices on its surface, which make it harder to achieve a good hygiene level when cleaning [17,35]. Practically, it is recommended to inspect and replace the rubber materials twice a year on a dairy farm to prevent the rubber from aging and becoming more porous [1]. It is shown in Table 6 that the optimized SAEW cleaning parameters needed for rubber gasket and PVC samples were similar. In order to simplify the operation to produce suitable SAEW, the treatment time, cleaning temperature, and ACC were optimized based on the models of rubber gasket and PVC samples with the levels needed to clean these two materials, which were 14.4 min, 29.6 °C, and 60 mg/L, respectively. These optimized SAEW parameters are applicable to dairy equipment with cold surfaces made with the same materials, such as piping systems, milk tanks, cryogenic milk trucks, and other equipment.

Three materials were cleaned using SAEW with the optimized parameters in order to verify the cleaning effects, which were compared with the results for the “super” treatment and its recommended solution properties (Figure 3). For SAEW, the bacteria and ATP removal rates were close to 100% for all three materials. Although the SAEW could not achieve the removal goals for bacteria and ATP in each treatment, it always had an equal or better effect as commercial detergents. This suggests that SAEW has potential as a replacement for typical commercial acidic chemical detergents for cleaning of milking systems. Meanwhile, the SAEW could be considered to be interchangeable use alkaline electrolyzed water in the CIP process, which would be environmentally friendly and highly advantageous to farmers [9,18].

## 4. Conclusions

This study simulated a worst case scenario for the CIP process in a milking system, with low flow when rinsing with warm water and washing with acidic solution, in order to check the cleaning effects of SAEW on stainless steel, rubber gasket, and PVC hose samples, involving laboratory trials with appropriate combinations of treatment time, water temperature, and ACC. Compared to chemical detergent, the SAEW showed significantly higher bactericidal properties in conventional CIP steps, which suggested its potential for being a cleaning and bacteria removal agent for milking systems. For stainless steel, the cleaning SAEW parameters were optimized at 9.9 min (treatment time), 37.8 °C (temperature), and 60 mg/L (ACC); and were 14.4 min, 29.6 °C, and 60 mg/L for rubber gasket and PVC samples, respectively. The cleaning effect and stability of SAEW need to be verified in a real CIP system in the future.

## Figures and Tables

**Figure 1 foods-09-01685-f001:**
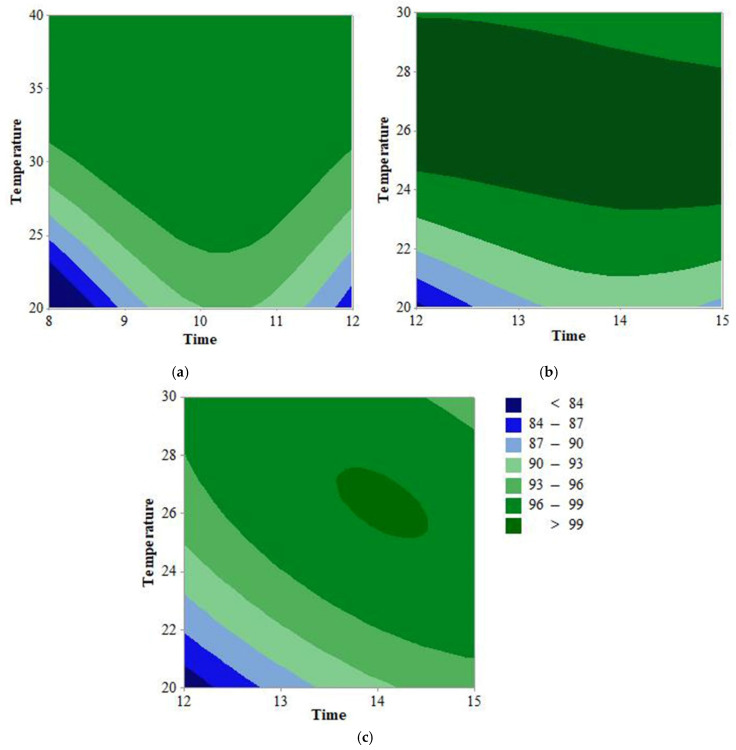
Response surface plots showing the effects of treatment time and cleaning temperature on stainless steel, rubber gasket, and PVC ATP removal rates (%) at an ACC of 50 mg/L. (**a**) Stainless steel; (**b**) Rubber gasket; (c) PVC.

**Figure 2 foods-09-01685-f002:**
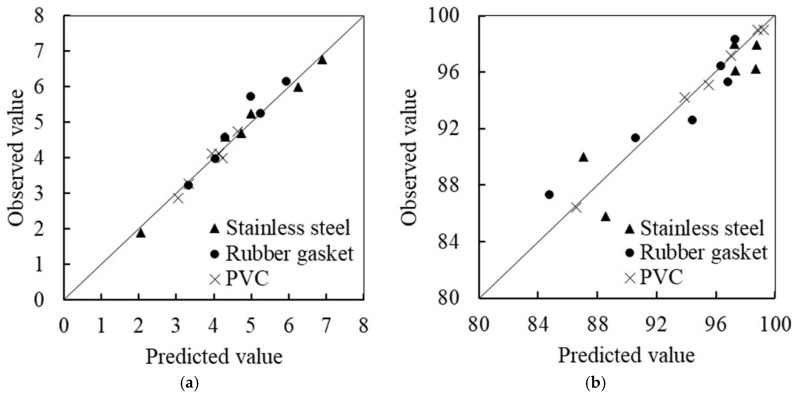
Observed and predicted bacterial reduction and ATP removal rates for three materials under 6 additional random experiments: (**a**) bacterial reduction (log_10_ CFU/cm^2^); (**b**) ATP removal rate (%).

**Figure 3 foods-09-01685-f003:**
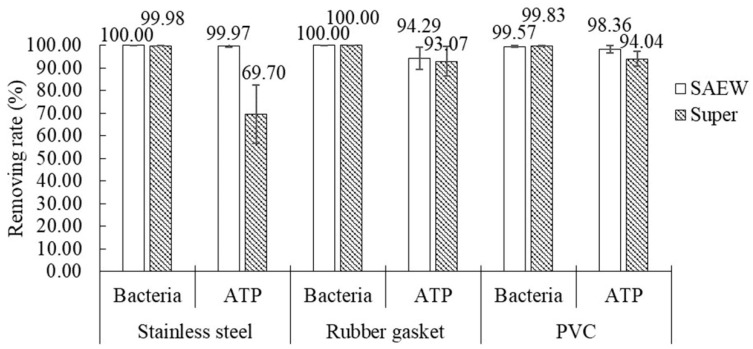
Comparison of bacteria and ATP removal rates for three materials after treatment with SAEW using the optimized parameters and commercial detergent (“super”). The required SAEW treatment time, cleaning temperature, and ACC were 9.9 min, 37.8 °C, and 60 mg/L to clean stainless steel; and were 14.4 min, 29.6 °C, and 60 mg/L to clean rubber gasket and PVC samples, respectively.

**Table 1 foods-09-01685-t001:** Box–Benken response surface design.

Trial	Time (min)				Temperature (°C)	ACC (mg/L)			
	SS ^1^	RG	PVC	SS	RG	PVC	SS	RG	PVC
1	10	15	15	40	30	30	40	50	50
2	8	15	15	30	25	25	40	60	60
3	10	12	12	30	20	20	50	50	50
4	8	12	12	40	25	25	50	60	60
5	8	15	15	30	25	25	60	40	40
6	8	13.5	13.5	20	25	25	50	50	50
7	12	12	12	30	30	30	60	50	50
8	12	13.5	13.5	40	30	30	50	40	40
9	10	15	15	40	20	20	60	50	50
10	10	13.5	13.5	30	20	20	50	40	40
11	12	13.5	13.5	30	25	25	40	50	50
12	10	13.5	13.5	20	20	20	40	60	60
13	12	12	12	20	25	25	50	40	40
14	10	13.5	13.5	20	30	30	60	60	60
15	10	13.5	13.5	30	25	25	50	50	50

^1^ SS = stainless steel pipe; RG = rubber gasket; PVC = PVC milk hose.

**Table 2 foods-09-01685-t002:** Six trials with different parameters used to validate the models.

Trial	Treatment Time (min)	Cleaning Temperature (°C)	ACC (mg/L)
1	9	20	40
2	9	30	50
3	9	40	60
4	11	20	60
5	11	30	50
6	11	40	40

**Table 3 foods-09-01685-t003:** Physicochemical properties of treatment solutions.

Solutions	ACC (mg/L)	pH	ORP (mV)
Tap water	0 ± 0	7.65 ± 0.01	402 ± 3
SAEW	18 ± 1	6.25 ± 0.04	732 ± 9
	30 ± 1	6.30 ± 0.07	814 ± 4
	42 ± 1	5.88 ± 0.05	904 ± 7
	53 ± 0	6.25 ± 0.11	928 ± 5
	62 ± 2	6.00 ± 0.02	924 ± 8
Super	0 ± 0	1.61 ± 0.01	692 ± 1

**Table 4 foods-09-01685-t004:** Bacteria and ATP removal from three materials with the SAEW and “super” treatments.

Solution	Trial	Removing Bacteria (log_10_ CFU/cm^2^)			Removing ATP (RLU)		
		SS ^1^	RG	PVC	SS	RG	PVC
SAEW	1	6.00 ± 0.00 *^b 2^	5.45 ± 0.00 *^a^	5.64 ± 0.00 *^b^	265 ± 0.00 ^b^	130 ± 2.12 ^b^	420 ± 0.71 ^b^
	2	2.86 ± 0.90 ^b 3^	4.23 ± 0.00 ^b^	3.18 ± 0.00 *^a^	225 ± 0.71 ^b^	231 ± 5.66 ^b^	405 ± 0.71 ^b^
	3	4.67 ± 0.38 ^a^	1.59 ± 0.34 ^b^	4.18 ± 1.07 ^b^	365 ± 0.00 ^b^	115 ± 0.00 ^b^	360 ± 2.12 ^b^
	4	6.00 ± 0.00 *^b^	1.79 ± 0.01 ^b^	3.18 ± 0.00 *^a^	265 ± 0.00 ^b^	224 ± 7.78 ^b^	399 ± 0.71 ^b^
	5	6.15 ± 0.00 *^b^	5.81 ± 0.00 *^b^	3.95 ± 0.00 *^b^	268 ± 1.41 ^b^	225 ± 0.71 ^b^	415 ± 2.83 ^b^
	6	2.50 ± 0.01 ^b^	2.11 ± 0.46 ^b^	3.02 ± 0.34 ^a^	120 ± 0.71 ^b^	170 ± 0.71 ^b^	415 ± 1.41 ^b^
	7	6.15 ± 0.00 *^b^	4.57 ± 1.75 ^a^	3.26 ± 0.00 ^a^	263 ± 8.49 ^b^	220 ± 2.83 ^a^	410 ± 2.12 ^b^
	8	6.00 ± 0.00 *^b^	5.56 ± 0.00 *^a^	3.98 ± 0.00 *^b^	270 ± 0.00 ^b^	185 ± 0.71 ^b^	325 ± 1.41 ^b^
	9	6.00 ± 0.00 *^b^	3.91 ± 0.51 ^b^	4.29 ± 1.91 ^b^	265 ± 0.00 ^b^	125 ± 0.71 ^b^	400 ± 0.00 ^b^
	10	4.67 ± 0.38 ^a^	2.94 ± 0.51 ^b^	3.98 ± 0.00 *^b^	355 ± 2.12 *^b^	165 ± 0.00 ^b^	310 ± 1.41 ^b^
	11	4.60 ± 0.71 ^a^	5.81 ± 0.00 *^b^	3.95 ± 0.00 *^b^	230 ± 0.00 ^b^	230 ± 0.71 ^b^	455 ± 0.71 ^b^
	12	2.95 ± 1.19 ^b^	5.44 ± 0.70 ^a^	4.29 ± 0.00 *^b^	215 ± 0.00 ^b^	275 ± 0.00 ^b^	303 ± 0.71 ^b^
	13	2.58 ± 0.44 ^b^	3.05 ± 0.91 ^b^	3.00 ± 0.00 *^a^	130 ± 0.71 ^b^	170 ± 0.71 ^b^	245 ± 0.00 ^b^
	14	4.48 ± 2.36 ^a^	5.94 ± 0.00 *^b^	4.29 ± 0.00 *^b^	267 ± 0.71 ^b^	285 ± 0.71 ^b^	313 ± 2.12 ^b^
	15	5.59 ± 0.49 ^b^	5.46 ± 0.50 ^a^	2.00 ± 1.41 ^b^	360 ± 1.41 ^b^	220 ± 2.12 ^a^	383 ± 1.41 ^b^
Super		4.18 ± 1.15 ^a^	5.06 ± 0.49 ^a^	2.83 ± 0.50 ^a^	92 ± 15.56 ^a^	219 ± 5.66 ^a^	392 ± 19.09 ^a^

^1^ SS = stainless steel pipe; RG = rubber gasket; PVC = PVC milk hose. ^2^ * No detectable survivors. ^3^ Different lowercase letter in each column indicate significant differences when comparing the items for “super” with the others at 0.05 level.

**Table 5 foods-09-01685-t005:** Response surface models for log_10_ bacterial reductions and ATP removal rates for three typical materials in a milking system cleaned using SAEW.

Materials	Removal Efficiency ^1^	Models ^2^	R^2^	*p*	Lack of Fit
Stainless steel	log_10_ bacterial reduction	R_S_ = −6.240 + 0.802x_1_ + 0.155x_2_ + 0.040x_3_ − 0.006x_1_x_3_ − 0.001x_2_x_3_ − 0.024x_1_^2^ − 0.001x_2_^2^ + 0.001x_3_^2^	0.90	0.05	0.24
	ATP removal rate	R_S-ATP_ = −84.900 + 23.800x_1_ + 3.590x_2_–1.153x_1_^2^ − 0.051x_2_^2^	0.70	0.01	0.14
Rubber gasket	log_10_ bacterial reduction	R_R_ = −86.500 + 12.120x_1_ + 0.191x_2_ − 0.423x_1_^2^	0.56	0.03	0.97
	ATP removal rate	R_R-ATP_ = −329 + 27.200x_1_ + 21.950x_2_ − 1.840x_3_ − 0.381x_1_x_2_ + 0.082x_1_x_3_ − 0.034x_2_x_3_ − 0.787x_1_^2^–0.285x_2_^2^ + 0.016x_3_^2^	0.94	0.01	0.57
PVC	log_10_ bacterial reduction	R_P_ = 51.000 − 1.605x_1_ − 3.123x_2_ + 0.076x_1_x_2_ + 0.042x_2_^2^	0.70	0.01	0.99
	ATP removal rate	R_P-ATP_ = −411 + 42.500x_1_ + 15.500x_2_ + 0.153x_3_ − 0.488x_1_x_2_ − 1.055x_1_^2^ − 0.164x_2_^2^	0.85	0.01	0.05

Note: ^1^ The unit of the log_10_ bacterial reduction model is log_10_ CFU/cm^2^, while that of the ATP removal rate model is %. ^2^ x_1_ is the treatment time in min; x_2_ is the temperature in °C; x_3_ is the ACC in mg/L.

**Table 6 foods-09-01685-t006:** Optimized parameters of SAEW treatments predicted by the models.

Material	Treatment Time (min)	Cleaning Temperature (°C)	ACC (mg/L)
Stainless steel pipe	9.9	37.8	60
Rubber gasket	13.8	28.5	40
PVC milk hose	14.9	30	60
Combining rubber and PVC	14.4	29.6	60

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
