# Peer review of "Optimization and Modeling of Slightly Acidic Electrolyzed Water for the Clean-in-Place Process in Milking Systems"

_foods, 2020, doi:10.3390/foods9111685_

Round 1

Reviewer 1 Report

This paper investigated optimization and modeling of slightly acidic  electrolyzed water for milking system clean-In-3 place process. The authors try to find an environmentally friendly and energy efficient alternative of acidic detergent in milking system clean-in-place process. This kind of study are important for environment and for human health.

My comments are listed below for the authors to consider:

  1. Innovation of this paper should be emphasized in the Introduction.

Line 19. 37.8 oC (water temperature), Please format 37.8 oC in correct way. In rest of article you have the same situation.

  1. Keywords: acidic electrolyzed water, response surface model, milking system, cleaning, disinfection In my opinion keywords should be differed from that in title.
  2. The title and content of the table must be self-explanatory, all abbreviations should be explained in the footer of the table.
  3. Table 2 lists the factors 1,2,3… I find that unfortunate and factors should be replaced.
  4. Line 132: 1 SS = stainless steel pipe; RG = rubber gasket; PVC = PVC milk hose. The same as below. What it meen: The same as below.
  5. In table 3 number must be precision (what it's meen?)
  6. In the methodology, please specify what specific statistical tools were used.
  7. In table 2 there is factor and in table 4 trial - it should be standardized
  8. There is no legend in Figure 2b
  9. In Table 6 you have material: Combining rubber and PVC, which has not been previously analyzed together.

Author Response

Dear reviewer,

  Thank you very much for the comments on our manuscript entitled "Optimization and modeling of slightly acidic electrolyzed water for milking system Clean-In-Place process". We have carefully considered the comments and made extensive revisions for our manuscript accordingly. The file of the point-by-point response was attached.

  Sincerely,

Chaoyuan Wang.

Reviewer 2 Report

The aim of this manuscript was to evaluate the efficacy of slightly acidic electrolyzed water (SAEW) to disinfect milking system made of stainless steel, gasket, and PVC milk hose in comparison to commercial acidic detergent. Overall the manuscript need some work to improve the structure. In addition, I have a few concerns expressed below:

Line 32 – 34: not clear, please rephrase

Line 41: Please rephrase = Studies showed that optimization of cleaning protocols aiming at..

Line 49: Please rephrase = EW could be disposed back into the environment after usage, contributing to decrease the high amount of halogenated compounds accumulating in the environment, as highlighted by the U.S. Environmental Protection Agency.

Line 56: Please rephrase = studies in the laboratory have shown that..

Line 58: Please rephrase = However, the practical challenge of using alkaline electrolyzed water and AEW is the higher corrosivity.

Line 59: have shown that..

Line 62: A study shown that..

Lie 64: Please rephrase = Besides, the consumption of water an energy to produce SAEW is 50% less than the one needed to produce the same quantity of AEW due to the different membrane structures.

Line 67: Please rephrase = however,  its effectiveness in cleaning milking systems has not been yet tested.

Line 68: Please rephrase = The successful application of SAEW alone in washing the milking system, would highly benefit the optimisation of the CIP cleaning protocols. Therefore, this work aimed at investigating..

Line 70: remove “by laboratory test”

Line 71: Please rephrase =  in addition, the aim of this study was to optimize the combinations of..

Line 76: please use Italic for the name of both bacteria

Line 77: remove “kinds of”

Line 85: Please rephrase = 1 mL of each culture broth was centrifuged (3K15, Sigma, Germany) at 4400×g for 4 min at 4 oC . Bacterial cells were resuspended in 10 mL of raw milk with a mixer

QUESTION - Line 93: what was the final concentration of bacteria inoculated on the surface of the specimens? Was the inoculation performed by dipping the speciment into a solution or by spreading?

Line 98: please provide the full name for ATP

QUESTION - Paragraph 2.4: were the bacteria inoculated singularly or as a cocktail? And how did you discriminated between the native flora in raw milk, E.coli and P. fluorescent on aerobic plate count plates? If the aim of spiking the sample was just to increase the bacterial count number in raw milk before treatment, why did you use both microorganisms?

Line 111: generator operating at what power?

Line 118: are displayed

Line 14: remove scientific

Table 2: what “ the same as below” refers to?

Paragraph 2.7 : Perhaps use as title only Disinfectant treatments. In addition, I think that having a small scheme with the standard CIP steps, showing the one that were eliminated in order to test your disinfection process will be beneficial to understand the experimental plan.

Line 138: convert reference to number style

Line 143: I would suggest to merge paragraph 2.8 with 2.6

Line 151: what type of statistical analysis were applied? Please specify.

Line 154: Great differences (statics?) were observed between the removing bacteria = please explain what this means. If this refers to the disinfection efficacy then please make the terminology clear across the text.

QUESTIONS about Table 4:

  • What was your initial concentration of bacteria on all the surfaces before disinfection? Would be helpful to have an additional row (untreated?) with this piece of information, or express the disinfection efficacy as a log reduction. If there were no detectable survivors, how was the number reported in the table calculated?

For example: in SS trial 11, the reported value of log 6.00 with no detectable survivors, means that the initial log count (before treatment) was 6.00 and all the bacteria (post treatment) were removed?

And also, in the case of PVC trial 2 (log 3.00,no survivors) and trial 15 (log 5.64, no survivors) how can this be? What was the initial inoculum in trial 2 to have no survivors? All the trials should start from the same log value.

Please clarify. 

  • What are the statistic applied? At which power were the results considered significant?

Line 155-162: Please rephrase =  it could be seen that in some condition (please specify which conditions you re referring to, trials n?), SAEW treatments had similar or higher bacteria (were this significant?) disinfection efficiency  as “Super", as shown by the ATP values and bacteria counts (table 4).  These results, indicate that using SAEW with appropriate parameter combinations of treatment time, water temperature and ACC has the potential to be a cleaning agent for milking system. The bacteria and ATP removing rates of stainless steel, rubber gasket, and PVC after using “Super” were 99.98%, 100.00%, 99.83%, and 69.70%, 93.07%, 94.04% = how was this percentage calculated?. While, after using SAEW with different combinations of treatment time, cleaning temperature, and ACC, could achieve the same or even better = what this means scientifically? Could you please express this in numbers/log reduction?

Line 172 - 174: Please rephrase = HClO can kill bacteria by destroying the membrane, leading to leakage of cytoplasmic content, proteins denaturation and stopping cellular metabolism.  Similarly, the disinfection efficiency of “Super” is based on H2SO4 and H3PO4, causing protein denaturation in the bacterial cell wall.

Line 176: again please define (give numbers) the use of “better”

Line 185: Please express as <0.05 or >0.05: which showed all models were logical=what this means?

Line 188: E. coli. Where the 0.3% bovine serum albumin comes from? Was not introduced anywhere in the manuscript.

Line 190 – 193: Please rephrase =  meaning not clear.

Table 5. Few comments

  • Design: instead than items maybe name the column as “Removal efficiency” and use Plate count and ATP rate as entries
  • Model: the R2 coefficient is actually pretty low for all the models. How do you justify this?

Line 205-206: significantly (how much?)

Line 212: Please Rephrase =The increase in temperature would result in more ACC releasing to air, and thus decreasing the sterilization effect of SAEW, especially when the temperature was over 45 oC.

Line 218 – 220: Please Rephrase = To validate the models, six additional random trials (Table 3) were carried out. The parameters selected were: treatment time from 8 to 12 min, temperature from 20 to 40 oC, and ACC from 40 to 60 mg/L for stainless steel.  Rubber gasket and PVC had a treatment time from 12 to 15 min, temperature from 220 20 to 30 oC, and ACC from 40 to 60 mg/L for.

Line 221: Please Rephrase = The results matched with the predicted..

QUESTION  Figure 3: how did you calculate the removing rate for bacteria? Please add the formula to the figure legend or in the material and methods.

Line 262: compared to

Line 263: SAEW achieved equivalent or better cleaning = quantify better

Author Response

(The authors gave the same response as above.)

Reviewer 3 Report

Overall, the manuscript by Liu et al. is well written. The authors optimized parameters of slightly acidic electrolyzed water (SAEW) alone treatment for three typical milking system materials (i.e., stainless steel (304) pipe, rubber gasket, and PVC milk hose) using response surface models. The findings are interesting and significant. However, several minor points need to be addressed or discussed.

  1. Why only bacteria and adenosine triphosphate were measured for the evaluation of performance of SAEW model? Thought it was mentioned in the discussion, this information should be added to the introduction part.
  2. For each material, only three specimens were used for bacteria and ATP measurements, respectively, for the original contaminating level and after cleaning. Did you do a power analysis before making a decision of using n = 3?
  3. Why only use ‘Super’ as a comparison treatment to SAEW? How about other treatments such as alkaline electrolyzed water (Alkaline EW) and alkaline chemical detergent? What is the novelty of current study compared to the previous study (reference 18)? How about the combination of SAEW and Alkaline EW? This should be discussed.
  4. How to transfer the application of SAEW from response surface models to real milking system?
  5. What are the limitations of current study? Most of the findings are descriptive and only Figure 3 showed real data of SAEW’s effectiveness when compared to commercial detergent (Super).

Author Response

(The authors gave the same response as above.)

Reviewer 4 Report

In this paper, authors well described the significant cleaning potential of the slightly acidic electrolyzed water. The results achieved are important in the point of the continuous growing awareness of the efficacy of possible alternatives to commercial chemical detergents that generally corrode and affect products quality. At the same time I really appreciated the effort of the authors to provide a ready to use protocol that can be easily followed by "on field" farmers.

However, there are few issues that should be addressed before publication:

- line 59: "Liu et al. 58 [18] verified that alkaline..:"

- Authors should better specify the kind of statistical analysis used to compare bacterial growth rather than describing the software used

- please check some referenceses that do not agree to journal guidelines

Author Response

(The authors gave the same response as above.)

Round 2

Reviewer 2 Report

The manuscript has been greatly improved and almost of all the comments were sufficiently addressed. Few minor comments as follow: 

  1. Line 21: after washing 
  2. Table 3: this is more part of the results than the methods. Consider to move it in a different section.
  3. Line 156 -162: this part does not really belong to the paragraph 2.8. I believe is more appropriate to move it to the section 2.7. Similarly, table 4 could be just included as additional material and the values reported as a range (min -max) in the section 2.1 and presented as " Initial concentration of bacteria were..."
  4. Line 199: clarify the meaning of "these of lack of fit were"
  5. Line 122; rephrase "this can explain why more time or higher ACC were needed to achieve satisfactory hygiene levels"

Author Response

Dear Reviewer,

    Thank you very much for the comments on our manuscript entitled "Optimization and modeling of slightly acidic electrolyzed water for milking system Clean-In-Place process". We have carefully considered the comments and made extensive revisions for our manuscript accordingly. The point-by-point responses are listed as an attachment.

    Sincerely,

Chaoyuan Wang.
